# DBS in Treatment of Post-Traumatic Stress Disorder

**DOI:** 10.3390/brainsci8010018

**Published:** 2018-01-20

**Authors:** Angelo Lavano, Giusy Guzzi, Attilio Della Torre, Serena Marianna Lavano, Raffaele Tiriolo, Giorgio Volpentesta

**Affiliations:** 1Unit of Functional and Stereotactic Neurosurgery/Operative Unit of Neurosurgery, Department of Medical and Surgical Sciences, University Magna Graecia of Catanzaro, Catanzaro 88100, Italy; guzzi.giusy@libero.it (G.G.); adt2005@tiscali.it (A.D.T.); raffaeletiriolo@live.it (R.T.); volpentestag@unicz.it (G.V.); 2Department of Health Science, University Magna Graecia of Catanzaro, Catanzaro 88100, Italy; serenalavano@yahoo.it

**Keywords:** posttraumatic stress disorder, deep brain stimulation, fear extinction, amygdala, prefrontal cortex

## Abstract

Post-traumatic stress disorder (PTSD) is a debilitating psychiatric condition for which pharmacological therapy is not always solvable. Various treatments have been suggested and deep brain stimulation (DBS) is currently under investigation for patients affected by PTSD. We review the neurocircuitry and up-to-date clinical concepts which are behind the use of DBS in posttraumatic stress disorder (PTSD). The role of DBS in treatment-refractory PTSD patients has been investigated relying on both preclinical and clinical studies. DBS for PTSD is in its preliminary phases and likely to provide hope for patients with medical refractory PTSD following the results of randomized controlled studies.

## 1. Introduction

In the Diagnostic and Statistical Manual of Mental Disorder–V (DSM-V) post-traumatic stress disorder (PTSD) is described not only as a feeling of fear and helplessness, as it was classified in the DSM-III, but also as a disorder including negative cognitions, negative emotional states, and reactivity symptoms [1]. Victims of sexual assault, serious accidents, sudden death of a loved one, or soldiers employed in war are exposed to various traumatic events with serious risk of developing PTSD, especially in the presence of comorbidity such as depression and substance abuse [2]. To make a diagnosis of PTSD, the patient must report the symptoms mentioned above for over 30 days following the traumatic event to an extent that it is an impairment in the quality of daily life. PTSD has significant economic and health repercussions; it is estimated prevalence in the United States is around 5–8% with a greater tendency in the female sex [3]. The risk of suicide attempt or ideation is also associated to PTSD [4]. Other risk factors involved in the pathophysiology of PTSD are genetic polymorphism [5], endocrine dysregulation [6], reduction of neurotrophic factors levels [7], as well as abnormal monoamine [8] and neuropeptide levels [9].

## 2. Materials and Methods

The anatomical structures involved in the neurocircuitry of fear conditioning are amygdala, prefrontal cortex, and hippocampus. Basolateral complex (BLA) of amygdala, consisting of the lateral nucleus (LA), the basal nucleus (BA), and the accessory basal nucleus, is the main receiver of the sensory afferences coming from two sources: the thalamus sensory nuclei and the primary sensory areas of the cerebral cortex. For many types of emotions, and especially for fear, the amygdala is of great importance, and valuable information retransmitted through this path reaches the amygdala more rapidly than sensory information retransmitted by the cortex. For example, lesions of the basolateral complex abolish classic fear conditioning [10]. Other nuclei of amygdala are the cortical nucleus, the central nucleus (CE), and intercalated cell clusters (ITC). The LA receives somatic, visual, and auditory sensory fibers, conveying a fast signal for danger [11,12] and it transmits the stimulus to the CE that is in connection with different hypothalamic and brainstem areas responsible for autonomic responses associated with fear [13]. Neurocircuitry of fear extinction differs from that of fear conditioning and it involves the ventromedial prefrontal cortex (vmPFC), the basolateral complex (BLA), the intercalated cell cluster (ITC) of the amygdala, and the hippocampus. The infralimbic (IL) subregion of medial prefrontal cortex (mPFC) is necessary for the inhibition of conditioned fear following extinction [14] and it drives extinction-related plasticity in the amygdala [15]. The BLA of amygdala is involved in extinction learning while vmPFC allows inhibition of fear during extinction recall [16]. In animal models, extinction may be due to an increased inhibition of fear output CE neurons related to an enhanced recruitment of GABAergic ITC cells by BLA inputs. It is likely that ITC neurons constitute mediators of extinction because they receive information about the conditioned stimulus from the basolateral amygdala (BLA), and contribute inhibitory projections to the CE, the main output station of the amygdala for conditioned fear responses [17]. Functional neuroimaging studies, based on single-photon emission tomography (SPECT), positron emission tomography (PET), and functional magnetic resonance imaging (fMRI) showed functional changes in the amygdala, hippocampus, and prefrontal cortex during emotional processing tasks and at rest in PTSD patients, consisting of increase of cerebral blood perfusion in the amygdala and decrease of perfusion in the superior frontal gyrus and parietal and temporal regions [18,19,20]. The dorsal anterior cingulate cortex, hippocampus, and insula appeared hyperactive [21] while simultaneous hypo-activity in the inferior occipital gyrus, ventromedial prefrontal cortex, rostral anterior cingulate cortex, para-hippocampal gyrus, lingual gyrus, dorsal amygdala and anterior hippocampus, orbitofrontal cortex, putamen, middle occipital gyrus, dorsomedial prefrontal cortex, dorsal anterior cingulate cortex and mid-cingulate appeared to be related to a greater symptom severity [22].

## 3. Results

Pharmacotherapy, based on paroxetine, sertraline, fluoxetine, risperidone, topiramate, and venlafaxine, associated with psychotherapy can be effective in PTSD [23]; however, many patients may be unresponsive or only partially responsive to medical treatment. It is unclear when a patient affected by PTSD can be considered treatment-resistant and if the coexistence of mental diseases or substance abuse are responsible for the refractoriness to conventional treatments [24]. The application of Deep Brain Stimulation (DBS) in PTSD is under investigation since the procedure has achieved promising results in the surgical treatment of other psychiatric disorders such as major depression and obsessive-compulsive disorder. The targets studied in preclinical models are the basolateral amygdala, ventral striatum, hippocampus, and prefrontal cortex. 

### 3.1. Basolateral Amygdala

Langevin et al. demonstrated a therapeutic response and a decrease of amygdala hyperactivity after DBS stimulation of the BLA complex in PTSD model rats using 4 h/die stimulation with 160 Hz, 120 μs and 2.5 Volts in monopolar configuration [25,26]. Development of epileptiform after discharge may be a potential side effect when high current intensities are used [27]. Based on these observations in 2014, a study was started on human clinical use of BLA DBS in six combat veterans affected by PTSD and it is still actively recruiting patients [28] Eligible subjects signed informed consent and underwent baseline evaluations for six-weeks with Clinician Administered PTSD Rating Scale (CAPS), baseline 18-FluoroDeoxyGlucose (18FDG) PET scan and additional baseline clinical evaluation of an independent team of psychiatrists and neurosurgeons. Patients with total CAPS score ≥85 at the end of baseline period could be kept in the study. The leads (Model 3387, Medtronic Minneapolis, USA) were implanted bilaterally in the BLA with standard precoronal trajectory and stimulation started four weeks after the implant with progressive increase of stimulation parameters to a maximum of 7 V, 200 Hz, and 210 μs [29,30]. A positive clinical response is defined as a 30% reduction in CAPS [31] from baseline and a Clinical Global Impression-Improvement (CGI-I) [32] score of 1 (very much improved) or 2 (much improved).

### 3.2. Ventral Striatum

The effects of ventral striatum (VS) DBS (100–200 μA, 0.1-ms pulse duration, 130 Hz) have been tested in a rodent PTSD model. DBS of the VS (the VC/VS homolog in rats) during extinction training allowed reduction of fear expression and strength of extinction memory; stimulation of dorso-medial VS, just above the anterior commissure, allowed facilitation of extinction while stimulation of more ventro-lateral sites in VS impaired extinction [33].

### 3.3. Hippocampus and Prefrontal Cortex

Disruptions of fear extinction-related potentiation of synaptic efficacy in the connection between the hippocampus (HPC) and the medial prefrontal cortex (mPFC) have been shown to impair the recall of memory extinction in rats. Moreover, low-frequency hippocampal stimulation delivered after extinction impaired the extinction learning and the development of hippocampal-PFC plasticity [34]. Medial Prefrontal Cortex DBS mitigated conditioned fear, partially improved anxiety-like behavior, and reduced BLA cell firing in a preclinical model of PTSD [35]. 

## 4. Conclusions

Treatment-resistant PTSD is an important mental health issue in terms of the number of people affected and morbidity and functional impairment associated with the disorder.

Neuroimaging studies in humans support the hypothesis of the involvement of limbic regions in the pathophysiology of the disorder. The application of DBS for PTSD is still strictly investigational and animal models suggest that stimulation of the amygdala, ventral striatum, hippocampus, and prefrontal cortex may be effective in fear extinction and anxiety-like behavior. The limited data from humans support the potential safety and effectiveness of high frequency DBS of the basolateral amygdala (BLA) in treating PTSD.

Optimal targets and stimulation parameters, greater knowledge of the action mechanisms, and established criteria of inclusion/exclusion need to be characterized prior to the launch of multidisciplinary larger scale studies, always keeping in mind the risks associated with the surgical procedure and long-term neurostimulation.

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
