# Peer review of "DBS in Treatment of Post-Traumatic Stress Disorder"

_brainsci, 2018, doi:10.3390/brainsci8010018_

Round 1

Reviewer 1 Report

The authors present an interesting review of the field of neuromodulation in PTSD. I think that the review is timely and the emphasis on neuroanatomy is appropriate. The authors will need to have the review proofread for English grammar.

In addition, on line 114: "six combat veterans were treated...". this sentence should be changed to reflect that this study is still ongoing as far as we are aware. A more accurate statement would be "six combat veterans are being recruited for treatment with BLA DBS..."

Author Response

I revised the highly similar sentences reducing the high level of overlap of your manuscript.
I have also moved some lines related to the DBS of the Basolateral amygdala within the text.

Reviewer 2 Report

The Authors provide updated and appropriate information about neuroanatomy and possible clinical relevance of DBS in the treatment of PTSD.

The Authors, report a very interesting review addressing a number of concepts involved in the field of neuromodulation of psychiatric disorders and, in particular, in the neurocircuitry that may be of relevance for the implementation of Deep Brain Stimulation (DBS) in posttraumatic stress disorder (PTSD). PTSD is a debilitating pathological condition associated with dysfunction in well-established neural circuits, including amygdala, ventral striatum, hippocampus, and prefrontal cortex. Although most patients improve with medications and/or psychotherapy, approximately 20–30% are considered to be refractory to conventional treatments. To date, there is a paucity of viable options for patients with refractory PTSD. As such, efforts have been made to investigate novel therapeutic options and, among these, DBS has shown promising results, above all in preclinical studies. Even if, several caveats need to be considered when data from animal models are to be translated, studies in animal models may help us understand mechanisms of DBS and improve the efficacy of this treatment. Moreover, the only clinical report published so far, a patient implanted with electrodes in the amygdala has shown striking improvements in PTSD symptom.

In our opinion, the Authors in the present review provide updated and appropriate information about neuroanatomy and possible clinical relevance of DBS in the treatment of PTSD.

For these reasons, we belive that the paper should be consider for publication, with minor revision.

It is necessary to correct the many errors, even grammatical, of english language.

Author Response

(The authors gave the same response as above.)
